# Three Clinical Clusters Identified through Hierarchical Cluster Analysis Using Initial Laboratory Findings in Korean Patients with Systemic Lupus Erythematosus

**DOI:** 10.3390/jcm11092406

**Published:** 2022-04-25

**Authors:** Ju-Yang Jung, Hyun-Young Lee, Eunyoung Lee, Hyoun-Ah Kim, Dukyong Yoon, Chang-Hee Suh

**Affiliations:** 1Department of Rheumatology, Ajou University School of Medicine, Suwon 16499, Korea; serinne20@hanmail.net (J.-Y.J.); nakhada@naver.com (H.-A.K.); 2Department of Biomedical Informatics, Ajou University School of Medicine, Suwon 16499, Korea; ajoustat@gmail.com (H.-Y.L.); e.angie.lee@gmail.com (E.L.); 3Center for Digital Health, Yongin Severance Hospital, Yonsei University Health System, Yongin 16995, Korea; yoon8302@gmail.com; 4Department of Molecular Science and Technology, Ajou University, Suwon 16499, Korea

**Keywords:** classification, cluster analysis, laboratory, linear discriminant analysis, systemic lupus erythematosus

## Abstract

Systemic lupus erythematosus (SLE) is a heterogeneous disorder with diverse clinical manifestations. This study classified patients by combining laboratory values at SLE diagnosis via hierarchical cluster analysis. Linear discriminant analysis was performed to construct a model for predicting clusters. Cluster analysis using data from 389 patients with SLE yielded three clusters with different laboratory characteristics. Cluster 1 had the youngest age at diagnosis and showed significantly lower lymphocyte and platelet counts and hemoglobin and complement levels and the highest erythrocyte sedimentation rate (ESR) and anti-double-stranded DNA (dsDNA) antibody level. Cluster 2 showed higher white blood cell (WBC), lymphocyte, and platelet counts and lower ESR and anti-dsDNA antibody level. Cluster 3 showed the highest anti-nuclear antibody titer and lower WBC and lymphocyte counts. Within approximately 171 months, Cluster 1 showed higher SLE Disease Activity Index scores and number of cumulative manifestations, including malar rash, alopecia, arthritis, and renal disease, than did Clusters 2 and 3. However, the damage index and mortality rate did not differ significantly between them. In conclusion, the cluster analysis using the initial laboratory findings of the patients with SLE identified three clusters. While disease activities, organ involvements, and management patterns differed between the clusters, damages and mortalities did not.

## 1. Introduction

Systemic lupus erythematosus (SLE) is a chronic autoimmune disease characterized by inflammatory responses in diverse organs due to an abnormal immune system, including autoantibody production or hyperactive immune cells [1]. It shows various manifestations depending on the organ in which the inflammatory response occurs, with varying severity, and treatment is determined by such manifestations. When patients have only a skin rash or mild arthritis, cytotoxic drugs are not indicated; however, when they have nephritis or vasculitis, aggressive treatment, including glucocorticoids and immunosuppressants, is necessary [2,3]. In addition, if inflammation is not well-controlled, sustained hyperactive immune responses can lead to organ damage, such as renal failure. Some patients have mild symptoms continuously, while other patients have recurrent episodes of flare-up or active disease.

The mortality and morbidity of SLE are still remarkable despite the fact that management of this disease has advanced over the past two decades [4,5]. The causes of mortality are serious infection, atherosclerosis, and active disease, and poor outcomes are associated with high disease activity and renal damage in patients with SLE [5,6,7]. Patients with higher disease activity are vulnerable to permanent organ damage owing to the need for glucocorticoids and immunosuppressants. These drugs play an essential role in controlling disease activity but result in complications, including infection and atherosclerosis, in patients with SLE [8,9]. Monitoring the current disease status and modifying treatment are essential to minimize organ damage and drug complications in the management of SLE [10]. Both clinical manifestations and laboratory findings should be used to monitor the disease status of patients within SLE. Several disease activity indices used to represent disease severity include the Systemic Lupus Erythematosus Disease Activity Index (SLEDAI) and British Isles Lupus Assessment Group Index (BILAG), which have been designed on the basis of clinical symptoms and laboratory findings [11,12,13,14]. These scoring systems have been used in clinical trials or research and are recommended to identify the degree of severity among different subsets of patients with SLE in clinical practice. Anti-double-stranded DNA (dsDNA) antibody and complement levels are used as disease activity biomarkers, with high anti-dsDNA antibody titers or low complement levels indicating high disease activity [15]. In addition, the Systemic Lupus International Collaborating Clinics (SLICC)/American College of Rheumatology (ACR) Damage Index could predict organ damage and mortality as a tool for evaluating the long-term outcomes of SLE [16]. These tools that identify disease activity could be used to classify patients with SLE, and such differentiation could help modify further treatment to prevent clinical worsening. Several attempts have been made to classify patients based on clinical or immunological data, suggesting different phenotypes of SLE [17,18,19]. The formulation of classified subtypes through more sophisticated statistical methods can help assess and manage the disease and educate patients with mixed symptoms.

Herein, we classified the phenotypic clusters of Korean patients with SLE using their initial laboratory findings at the time of SLE diagnosis. Classifying clusters of SLE aimed to analyze subgroup characteristics, including their disease presentation and activities, management patterns, organ damage, and mortality.

## 2. Materials and Methods

### 2.1. Study Design

A total of 389 patients who met the Systemic Lupus International Collaborating Clinics (SLICC) classification criteria and the revised ACR classification criteria for SLE receiving standard-of-care treatment for SLE were enrolled [20,21]. The laboratory findings obtained at the time of SLE diagnosis, including complete blood count, erythrocyte sedimentation rate (ESR), C-reactive protein (CRP) level, complement 3 (C3) and 4 (C4) levels, anti-nuclear antibody (ANA) titer, and anti-dsDNA antibody level, were collected. The ANA titer was measured via immunofluorescence assay using ANA HEp-2 Plus (GA Generic Assays, Dahlewitz, Germany) and categorized by the International Consensus on ANA Patterns (ICAP) as follows: homogenous (AC-1), speckled (AC4,5), nucleolar (AC-8,9,10), or cytoplasmic (AC-15 to AC-23) [22]. The anti-dsDNA antibody level was also measured via enzyme immunoassay (GA Generic Assays), with a cut-off value of 7 IU/mL. The duration was defined as the period from the time when the initial tests were performed to the time when the cumulative data were collected. Data on cumulative manifestations, including oral ulcer, malar rash, alopecia, arthritis, and renal disease, were obtained. Disease activity and disease-related damage were assessed using the SLEDAI score and SLICC/ACR damage index at the time of data collection [16,23]. Comprehensive medication histories, including the use of glucocorticoids and immunosuppressants, were obtained. For medication data for immunosuppressants, taking the drugs for more than 1 month was considered as “use”. Data were collected from the medical records within 9 years (2006–2015) of patients with SLE managed at Ajou University Hospital using MS-SQL 2012 (Microsoft, Redmond, WA, USA). This study was conducted in accordance with the principles of the Declaration of Helsinki. The study protocol was reviewed and approved by the institutional review board of Ajou University Hospital (AJIRB-MED-MDB-17-147); the need for informed consent was waived because of the retrospective nature of the study.

### 2.2. Statistical Analysis

Hierarchical cluster analysis was performed on 389 patients with SLE with 10 different laboratory values. The laboratory values, including the white blood cell (WBC), lymphocyte, and platelet counts; hemoglobin, CRP, C3, C4, and anti-dsDNA antibody levels; ESR; and ANA titer, were transformed into Z-scores for hierarchical clustering [24]. To classify the patients with SLE according to laboratory values, we applied Ward’s method as an agglomeration method applied with Spearman correlation as a distance metric [25]. The clinical characteristics among clusters were examined using ANOVA with Tukey’s and Fisher’s exact tests (SPSS version 23.0; IBM SPSS Statistics, IBM Corporation, Chicago, IL, USA). To predict the classified SLE clusters, we constructed a discriminant model using Fisher’s linear discriminant (LD) functions with six variables, including the WBC count, ESR, C3 level, C4 level, anti-dsDNA antibody level, and ANA titer. Statistical significance was set at *p* < 0.05.

## 3. Results

### 3.1. Three Clusters with Different Characteristics Were Identified among the Patients with SLE

A total of 389 patients with SLE were divided into three clusters via hierarchical cluster analysis based on the 10 laboratory values (Figure 1). The analysis showed different patterns between the following two laboratory sets: positive sign set for SLE, including the ESR, CRP level, anti-dsDNA antibody level, and ANA titer, and negative sign set, including the C3 level, C4 level, hemoglobin level, platelet count, WBC count, and lymphocyte counts. In contrast, Cluster 2 showed a tendency to deviate from Cluster 3 in terms of the ANA titer, platelet count, WBC count, and lymphocyte count.

Table 1 shows the differences between the initial laboratory test results. The patients in Cluster 1 were significantly younger than those in the other clusters (1 vs. 2, *p* = 0.044 and 1 vs. 3, *p* < 0.001). Sex and disease duration did not differ between the SLE clusters. Cluster 1 had significantly higher anti-dsDNA antibody titers and lower platelet counts, hemoglobin levels, and C3/4 levels than Clusters 2 and 3 (*p* < 0.001, *p* = 0.004, *p* < 0.001, and *p* < 0.001, respectively). Cluster 2 had a significantly higher lymphocyte count and a lower ESR (*p* < 0.001 and *p* < 0.001, respectively). The WBC count was significantly lower in Cluster 3 than in the other clusters (3 vs. 1, *p* = 0.029 and 3 vs. 2, *p* < 0.001, respectively). The ANA titers were significantly different between all SLE clusters (*p* < 0.001), with the highest values in Cluster 3, followed by those in Clusters 1 and 2. The proportion of the homogeneous type (AC-1) was significantly higher in Clusters 1 and 2 (46/131, 35.4% and 64/183, 35%, respectively, *p* = 0.002) than in Cluster 3. The proportion of the nucleolar (AC-8,9,10) and cytoplasmic types (AC-15 to AC-23) was significantly higher in Cluster 2 (15/183, 8.2% and 22/183, 12.0%, respectively, *p* < 0.001) than in Clusters 1 and 3. The proportion of the speckled type (AC-4,5) was significantly different between all SLE clusters, with the highest proportion in Cluster 3 (54/75, 72%), followed by those in Clusters 1 and 2 (46/131, 44.6% and 56/183, 30.6%, respectively, *p* < 0.001). There was no significant association between the CRP level and SLE clusters.


### 3.2. Three Clusters Were Separated with Significant Statistical Power

An LD analysis (LDA) classification model was developed to identify the patients with SLE and assign them to one of the three clusters. The clusters were classified significantly according to the variables and LD1, LD2, and LD3, and the values were the coefficients of each parameter (Table 2).

Each patient’s value was identified in the constructed model of canonical discriminant functions (Figure 2). The canonical plot shows that the clusters were separated with an accuracy of 84.5% (72.5% in Cluster 1, 85.3% in Cluster 2, and 92.9% in Cluster 3). Clusters 1 and 3 showed distinctly different positions, and Cluster 2 was found in the intermediate region. This finding indicates that each cluster was characterized by the initial laboratory values that were sufficiently distinctive to allow the construction of discriminator segregating subgroups.

### 3.3. Each Cluster Showed Different Manifestations during Follow-Up

The manifestations in each cluster from the time of classification to the time of collection of the medical records were compared (Table 3). Cluster 1 had a higher number of clinical manifestations than Clusters 2 and 3 (*p* = 0.002 and *p* < 0.001, respectively). Further-more, Cluster 1 showed the highest prevalence of malar rash, alopecia, renal disease, azathioprine and cyclophosphamide use, and glucocorticoid use (*p* = 0.006, *p* = 0.001, *p* < 0.001, *p* = 0.004, and *p* < 0.001, respectively); however, the prevalence of oral ulcers was significantly higher in Cluster 2 than in Cluster 1 (*p* = 0.042). The prevalence of arthritis and serositis did not significantly differ between the SLE clusters; however, arthritis had an overall high prevalence in all clusters.

The SLEDAI score, which was calculated at the time of enrollment, was higher in Cluster 1 (7.2 ± 4.9) than in Clusters 2 (3.0 ± 3.2, *p* < 0.001) and 3 (2.4 ± 2.7, *p* < 0.001), while the SLICC/ACR damage index, which was also collected and calculated at the time of enrollment, did not differ. The proportion of patients taking hydroxychloroquine (HCQ) was lower in Cluster 1 (61.8%) than in Clusters 2 (75.4%, *p* = 0.013) and 3 (76.0%, *p* = 0.045). Although the proportion of patients currently taking glucocorticoids was similar in the three clusters, the total and mean doses of glucocorticoids were significantly higher in Cluster 1 than in Cluster 2 (*p* = 0.008 and *p* = 0.001, respectively). The patients in Cluster 1 took azathioprine more frequently than did those in Cluster 3 (*p* = 0.001) and cyclophosphamide more frequently than did those in Clusters 2 and 3 (*p* < 0.001 and *p* = 0.019, respectively).

### 3.4. Mortality and Renal Damage within 171 Months were Not Different between the Clusters

Fifteen patients (3.9%) died, while five patients (1.3%) had progressed to end-stage renal disease (ESRD). There was no difference in the mortality rate during the average follow-up period of 171 months between the clusters (data not shown).

## 4. Discussion

The cluster analysis using laboratory findings that were obtained at the time of SLE diagnosis identified three clusters among patients with SLE. Each cluster shared similar clinical characteristics, including laboratory findings and manifestations. These data allowed the determination of one of the subtypes using LDA based on the initial WBC count, ESR, C3 and C4 levels, anti-dsDNA antibody level, and ANA titers. The LDA classification model demonstrated a high level of spectral discrimination (84.5%), which reflected the utility of the serum levels of inflammatory or autoimmune markers and the homogeneity of patients in each cluster. It is also evident that the three clusters can be considered separately in patients with SLE, and this model can be helpful in understanding the clinical features of patients with SLE in clinical practice.

Several studies have attempted to classify patients with SLE using clinical or immunological features. One study showed two subsets of patients: Most patients in the active disease subgroup were of Black African descent and were diagnosed when younger, while the patients in the other sub-group had different features [26]. In a recent study, four discrete clusters were identified on the basis of patients’ symptoms, while disease characteristics, patient-reported outcomes, and treatment received in each cluster were significantly different [17]. A K-means cluster analysis based on patterns of clinical symptoms and mortality identified three clusters, and the cluster with frequent renal and hematologic symptoms showed a higher mortality in Chinese patients with SLE [27]. A cohort of Spanish patients with SLE showed that cardiovascular and musculoskeletal damage among several damage patterns was correlated with mortality [28]. An identification of three clusters using the damage index scale concluded that the cluster with prevalent renal and ocular damage had the highest damage score [19].

In this study, laboratory findings obtained at the time of SLE diagnosis were used to identify the three clusters. The levels of complement and anti-dsDNA antibodies in Cluster 1 were significantly different from those in Clusters 2 and 3. As both abnormal levels were included in the SLEDAI score, the score was also elevated in Cluster 1. Cluster 1 represented patients with SLE with a high disease activity who received a higher dose of glucocorticoids. High levels of anti-dsDNA antibody or low levels of complements indicate an active disease and predict a poor prognosis in patients with SLE [29,30]. Both markers are known to correlate with activity of lupus nephritis (LN) [31].

To summarize the results of clinical manifestations of clusters briefly, cluster 1 had higher numbers of clinical manifestations and SLEDAI scores and malar rash and renal disease more frequently and showed less frequent use of hydroxychloroquine and more frequent use of cyclophosphamide than Cluster 2 or 3. In addition, Cluster 1 had oral ulcer and alopecia more frequently, showed more frequent use of azathioprine, and took higher doses of glucocorticoids than Cluster 2. Clusters 2 and 3 had no difference in clinical manifestations.

In general, most patients with SLE are diagnosed in their 30s, and patients who develop SLE at a younger age have an active disease [26,32]. The diagnostic age was the lowest in Cluster 1, which represents the active disease group. However, the WBC count was not lower in Cluster 1, and the lymphocyte count was lower in Cluster 3, while the hemoglobin level was lower in Cluster 1 than in the other clusters. Cytopenia is the main hematologic manifestation in most SLE classification criteria and is derived from destruction to autoimmune response in patients with SLE [33]. SLE patients with lymphopenia have reduced surface expression of complement regulatory proteins and endogenous production of type 1 interferon [34]. An analysis of autoimmune cytopenia before or at childhood-onset SLE showed that patients with autoimmune cytopenia had a lower incidence of arthritis and a lower 2-year incidence of LN than those without autoimmune cytopenia [35]. Herein, cytopenia is not a typical finding of a particular cluster and might occur through the different etiologies of arthritis and renal involvement in SLE.

The presence of ANA is a typical feature of SLE, and its types and titers vary and have distinct roles, including non-pathological and pathogenic roles [36,37]. As a diagnostic marker for SLE, the ANA titer is known to have a sensitivity of 93% and a specificity of 57%. The titer of ANA and disease status of SLE are not regarded as relevant [38]. A high ANA titer does not indicate active inflammation, while a low ANA has not been ignored. Cluster 3 (mild disease) had the highest ANA titers and the lowest WBC and lymphocyte counts, while the ANA titer was not higher in Cluster 1 (active disease).

More than 10 types of ANA patterns have been reported and categorized into nuclear, cytoplasmic, and cell-cycle-related types. A recent study revealed that 36.5% of checked ANAs were homogeneous (AC-1), 19.9% speckled (AC-4,5), and 17.0% nucleolar (AC-8,9,10) among 9268 patients with positive ANAs [39]. The speckled type is known to be more specific for the diagnosis of SLE; however, its association with disease activity has not been identified. Among the several types of ANA, the homogeneous type was more frequently observed in Clusters 1 and 2 than in Cluster 3 and the speckled type in Clusters 1 and 3 than in Cluster 2 in this study. The speckled type might not be typical in patients with active SLE but was associated with the characteristics of Cluster 3.

Interestingly, the proportion of patients taking HCQ and the doses of glucocorticoids in Cluster 1 differed from those in Clusters 2 and 3. These results suggest that a lower proportion of patients taking HCQ might have a higher disease activity and might receive higher doses of glucocorticoids in Cluster 1. HCQ has been known to prevent disease flare-up or severe manifestations, including LN, and its maintenance has been associated with better prognosis in patients with SLE [40,41]. Some patients could not maintain HCQ owing to adverse effects, including retinopathy, or refused to take the medicine. Our data confirmed that the discontinuation of HCQ could be associated with disease activation in patients with SLE.

A limitation of this study is that there may be biases that arise from research methods that use data collected retrospectively. The data were dependent on medical records, and the follow-up time differed among the patients. In addition, biologic drugs, including belimumab and rituximab, were not included because they were not available for patients with SLE in Korea during the data collection period (2006–2015). The disease status of SLE is changing, and the SLEDAI score was collected only once. The number of cumulative manifestations and SLICC/ACR damage index were included to compensate for such weak points. However, our study suggests that patients with SLE can be classified into three subgroups based on the initial laboratory findings at the time of SLE diagnosis. Since each subgroup herein had different clinical characteristics, clinicians need to consider which subgroup of patients should be included for further management, and clinical trials should be designed to categorize which subgroups the study should enroll in.

## 5. Conclusions

In conclusion, the cluster analysis using the initial laboratory findings divided the patients with SLE into three clusters showing a clear differences in clinical symptoms and drug history. Cluster 1, which had the highest disease activity markers at SLE diagnosis, had a higher number of clinical manifestations within approximately 10 years than Clusters 2 and 3. However, prognosis, including mortality or ESRD, did not differ between the clusters.

## Figures and Tables

**Figure 1 jcm-11-02406-f001:**
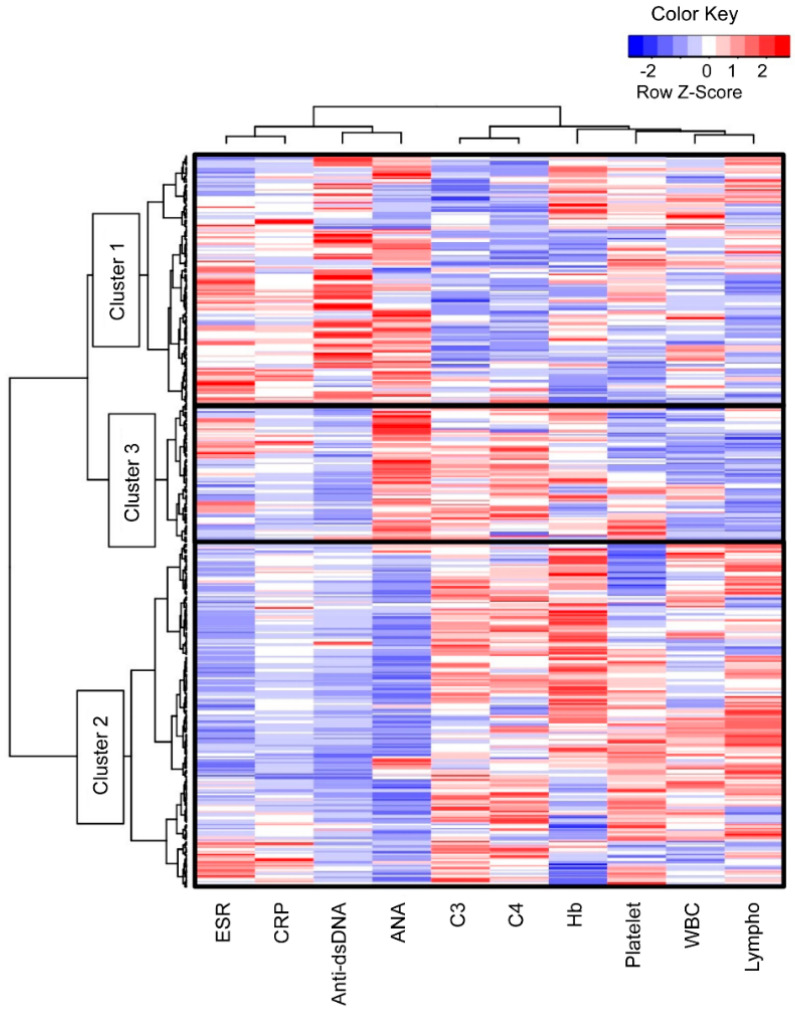
Three subgroups of patients with SLE divided via hierarchically cluster analysis. The patients with SLE (*n* = 389) were divided into three clusters based on the laboratory values at the time of SLE diagnosis. ANA, anti-nuclear antibody; C3, complement 3; C4, complement 4; CRP, C-reactive protein; dsDNA, double-stranded DNA; ESR, erythrocyte sedimentation rate; Hb, hemoglobin, lympho, lymphocyte; SLE, systemic lupus erythematosus; WBC, white blood cell.

**Figure 2 jcm-11-02406-f002:**
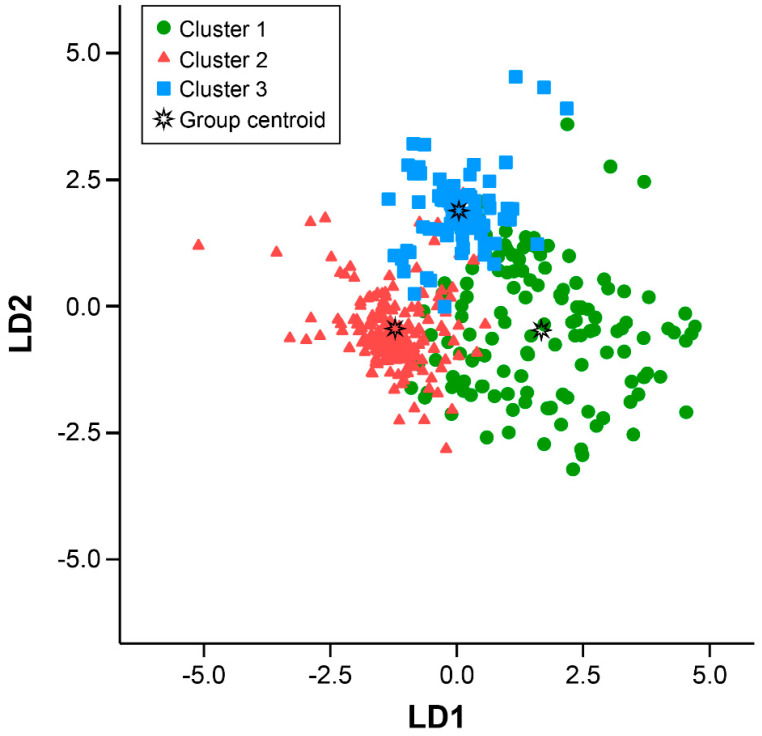
Three subgroups of patients with SLE identified at an accuracy of 84.5%. Canonical discriminant function shows that the LD was 72.5% in Cluster 1, 85.3% in Cluster 2, and 92.9% in Cluster 3. LD, linear discriminant; SLE, systemic lupus erythematosus.

**Table 1 jcm-11-02406-t001:** Clinical characteristics of the clusters according to the laboratory findings at the time of SLE diagnosis.

	Cluster 1	Cluster 2	Cluster 3	*p*-Value ^a^			
	(*n* = 131)	(*n* = 183)	(*n* = 75)	Overall	1 vs. 2	1 vs. 3	2 vs. 3
Diagnostic age, years	31.2 ± 13.2	35.6 ± 12.6	36.8 ± 12.3	<0.001	0.044	<0.001	0.752
Male: female, *n* (%)	10 (7.6):121 (92.4)	15 (8.2):168 (91.8)	4 (5.3):71 (94.7)	0.819	1.000	0.775	0.601
Duration, month ^b^	117.8 ± 48.5	126.0 ± 40.6	138.4 ± 146.8	0.169	0.608	0.144	0.455
WBC count,/µL	5623.9 ± 3157.4	6111.9 ± 2304.6	4673.1 ± 1877.5	<0.001	0.219	0.029	<0.001
Lymphocyte count,/µL	1214.3 ± 628.1	1765.5 ± 683.0	1171.2 ± 395.7	<0.001	<0.001	0.881	<0.001
Hemoglobin,/µL	11.5 ± 1.8	12.5 ± 1.6	12.3 ± 1.1	<0.001	<0.001	0.001	0.581
Platelet count, ×10^3^/µL	208.1 ± 76.7	238.9 ± 88	225.7 ± 66.7	0.004	0.003	0.288	0.452
ESR, mm/h	30.4 ± 26.5	16.8 ± 17.9	26.7 ± 20.1	<0.001	<0.001	0.454	0.003
CRP, mg/dL	1.1 ± 3	0.5 ± 1.8	0.7 ± 2	0.097	0.078	0.554	0.756
Complement 3, mg/dL	71.8 ± 29.4	112.7 ± 27.3	110.5 ± 21.7	<0.001	<0.001	<0.001	0.820
Complement 4, mg/dL	13 ± 7.5	25 ± 9.7	27.5 ± 8.6	<0.001	<0.001	<0.001	0.100
Anti-dsDNA antibody, IU/mL	47.8 ± 38.5	7.2 ± 10.3	7.5 ± 6.8	<0.001	<0.001	<0.001	0.996
ANA titer	1715.1 ± 1135.3	463.8 ± 641.6	2474.7 ± 731.5	<0.001	<0.001	<0.001	<0.001
Homogenous (AC-1), *n* (%)	46 (35.4)	64 (35)	11 (14.7)	0.002	1.000	0.001	0.001
Nucleolar (AC-8,9,10), *n* (%)	0 (0)	15 (8.2)	1 (1.3)	<0.001	<0.001	0.366	0.045
Speckled (AC-4,5), *n* (%)	58 (44.6)	56 (30.6)	54 (72)	<0.001	0.012	<0.001	<0.001
Cytoplasmic (AC-15 to AC-23), *n* (%)	3 (2.3)	22 (12)	1 (1.3)	<0.001	0.001	1.000	0.004
Mixed, *n* (%)	21 (16.2)	26 (14.2)	8 (10.7)	0.575	0.634	0.306	0.545

ANA, anti-nuclear antibody; CRP, C-reactive protein; dsDNA, double-stranded DNA; ESR, erythrocyte sedimentation rate; NA, not available; SLE, systemic lupus erythematosus; WBC, white blood cell. ^a^
*p*-Values were calculated using ANOVA with Tukey’s and Fisher’s exact tests. ^b^ Duration was defined as the period between the initial test and cumulative data collection. Continuous variables are presented as means ± standard deviations.

**Table 2 jcm-11-02406-t002:** Fisher’s linear discriminant functions for clustering.

Parameters	LD1	LD2	LD3
White blood cell count	0.001	0.001	0.000
Erythrocyte sedimentation rate	0.013	−0.040	−0.014
Complement 3	0.085	0.138	0.115
Complement 4	0.000	0.086	0.174
Anti-dsDNA Ab	0.103	0.041	0.028
Anti-nuclear antibody level	0.002	0.000	0.003
Constant	−10.259	−11.298	−14.753

LD, linear discriminant.

**Table 3 jcm-11-02406-t003:** Cumulative manifestations and treatment patterns of the SLE clusters.

	Cluster 1	Cluster 2	Cluster 3	*p*-Value ^a^			
	(*n* = 131)	(*n* = 183)	(*n* = 75)	Overall	1 vs. 2	1 vs. 3	2 vs. 3
Number of CMs, *n* (%)	1.4 ± 1.3	0.9 ± 0.9	0.8 ± 1	<0.001	<0.001	0.002	0.901
Number of CMs of ≥ 2, *n* (%)	49 (37.7)	41 (22.4)	16 (21.3)	0.006	0.004	0.019	1.000
Oral ulcer, *n* (%)	18 (13.7)	43 (23.5)	15 (20)	0.095	0.042	0.244	0.624
Malar rash, *n* (%)	35 (26.7)	27 (14.8)	8 (10.7)	0.006	0.010	0.007	0.430
Alopecia, *n* (%)	35 (26.9)	19 (10.4)	12 (16)	0.001	<0.001	0.085	0.212
Arthritis, *n* (%)	42 (32.1)	52 (28.4)	16 (21.3)	0.263	0.533	0.110	0.278
Renal disease, *n* (%)	46 (35.1)	24 (13.1)	12 (16)	<0.001	<0.001	0.004	0.556
Serositis, *n* (%)	2 (1.5)	0 (0.0)	0 (0)	0.150	0.173	0.535	NA
SLEDAI score *	7.2 ± 4.9	3.0 ± 3.2	2.4 ± 2.7	<0.001	<0.001	<0.001	0.548
SLICC/ACR damage index	0.4 ± 0.9	0.4 ± 1.0	0.4 ± 0.9	0.993	0.992	0.999	0.998
Hydroxychloroquine use, *n* (%)	81 (61.8)	138 (75.4)	57 (76)	0.021	0.013	0.045	1.000
Current glucocorticoid use, *n* (%)	120 (91.6)	159 (86.9)	65 (86.7)	0.385	0.208	0.339	1.000
Total glucocorticoid dose, mg	8465.9 ± 10,962	5306.0 ± 8645.4	5611.1 ± 6466.9	0.008	0.008	0.080	0.968
Mean glucocorticoid dose, mg	67.1 ± 76	37.8 ± 54.1	51.5 ± 82.3	0.001	0.001	0.254	0.305
Azathioprine use, *n* (%)	35 (26.7)	22 (12)	14 (18.7)	0.004	0.001	0.235	0.170
Cyclophosphamide use, *n* (%)	19 (14.5)	4 (2.2)	3 (4)	<0.001	<0.001	0.019	0.418
MMF use, *n* (%)	17 (13)	12 (6.6)	5 (6.7)	0.127	0.074	0.240	1.000
Methotrexate use, *n* (%)	14 (10.7)	31 (16.9)	13 (17.3)	0.243	0.142	0.200	1.000

CM, cumulative manifestation; MMF, mycophenolate mofetil; SLE, systemic lupus erythematosus; SLEDAI, Systemic Lupus Erythematosus Disease Activity Index; SLICC/ACR, Systemic Lupus International Collaborating Clinics/American College of Rheumatology. ^a^
*p*-Values were calculated using ANOVA with Tukey’s and Fisher’s exact tests. Continuous variables are presented as means ± standard deviations. * Most recent clinical visit.

## Data Availability

The data presented in this study are available on request from the corresponding author.

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
