# Peer review of "Three Clinical Clusters Identified through Hierarchical Cluster Analysis Using Initial Laboratory Findings in Korean Patients with Systemic Lupus Erythematosus"

_jcm, 2022, doi:10.3390/jcm11092406_

Round 1

Reviewer 1 Report

In the manuscript the comprehensive analysis of different clinical and laboratory findings in SLE patients was performed. Patients were subdivided into three clusters, which shared similar clinical characteristics. Authors confirmed the great diversity among SLE symptoms and identified factors which correlated with the high activity of the disease. The paper is original, relevant and interesting but the results do not surprise. The paper is well written, clear and easy to read and pointed out the great number of cumulative manifestations which can influence disease activity.

Author Response

Thank you for your valuable comment. 

Reviewer 2 Report

This is an interesting retrospecyive clinical evaluation of SLE course in a Korean population. In my opinion, major drowbacks of the study are as follows:

  1. Conclusions from the obtained results should be formulated. They are lacking in the abstract and the results are repeated at the end of the discussion instead of summarising the meaning and the importance of findings.
  2. I think that also the study aim should be better explained. This could help in drawing conclusions. If the only aim was to classify and the classification was performed, the only conclusion can be that the classification was performed. Why did the authors classify SLE patients? I think that classification was the tool, not the aim.

Author Response

We appreciate your review of our manuscript “Three Clinical Clusters Identified Through Hierarchical Cluster Analysis Using Initial Laboratory Findings in Korean Patients with Systemic Lupus Erythematosus”. In response to your comments and those of the reviewers, we have made several changes to the text, as summarized below.

  1. Conclusions from the obtained results should be formulated. They are lacking in the abstract and the results are repeated at the end of the discussion instead of summarising the meaning and the importance of findings.

Answer) Thank you for your valuable comment. We corrected the sentences in the Abstract and the Discussion, and underlined in the revised manuscript as below.

  • However, the damage index and mortality rate did not differ significantly between them. In conclusion, the cluster analysis using the initial laboratory findings of the patients with SLE identified 3 clusters. While disease activities, organ involvements, and management patterns differed be-tween the clusters, damages and mortalities did not.
  • In general, most patients with SLE are diagnosed in their 30s, and patients who develop SLE at a younger age have an active disease [25,31]. The diagnostic age was the lowest in Cluster 1, which represents the active disease group. However, the WBC count was not lower in Cluster 1, the lymphocyte count was lower in Cluster 3, while the hemoglobin level was lower in Cluster 1 than in the other clusters.
  • Herein, cytopenia is not a typical finding of a particular cluster, and might be occurred through the different etiologies of arthritis and renal involvement in SLE.
  • Cluster 3 (mild disease) had the highest ANA titers and the lowest WBC and lymphocyte counts, while the ANA titer was not higher in Cluster 1 (active disease).
  • In conclusion, the cluster analysis using the initial laboratory findings divided the patients with SLE into three clusters showing a clear differences in clinical symptoms and drug history. Cluster 1, which had the highest disease activity markers at SLE diagnosis, had a higher number of clinical manifestations within approximately 10 years than Clusters 2 and 3. However, prognosis including mortality or ESRD did not differ between the clusters.
  1. I think that also the study aim should be better explained. This could help in drawing conclusions. If the only aim was to classify and the classification was performed, the only conclusion can be that the classification was performed. Why did the authors classify SLE patients? I think that classification was the tool, not the aim.

Answer) Thank you for your valuable comment. We corrected the study aim in the Introduction, and underlined in the revised manuscript as below.

  • Herein, we classified the phenotypic clusters of Korean patients with SLE using their initial laboratory findings at the time of SLE diagnosis. Classifying clusters of SLE aimed to analyze subgroup characteristics including their disease presentation and activities, management patterns, organ damage, and mortality.
  • The cluster analysis using laboratory findings which were obtained at the time of SLE diagnosis identified three clusters among patients with SLE. Each cluster shared similar clinical characteristics, including laboratory findings and manifestations.

Thank you for the constructive review. We hope that the revised manuscript now meets the journal’s standards for publication.